# The Impact of COVID-19 Confinement on Substance Use and Mental Health in Portuguese Higher Education Students

**DOI:** 10.3390/healthcare11040619

**Published:** 2023-02-19

**Authors:** Ana Paula Oliveira, Henrique Luis, Luís Soares Luís, Joana Rita Nobre, Lara Guedes Pinho, Núria Albacar-Riobóo, Carlos Sequeira

**Affiliations:** 1Health School, Polytechnic Institute of Portalegre, 7300-555 Portalegre, Portugal; 2Faculty of Nursing, University of Rovira e Virgili, 43003 Tarragona, Spain; 3Unidade de Investigação em Ciências Orais e Biomédicas (UICOB), RHODes—Dental Hygienists for Sciences, Faculdade de Medicina Dentária, Universidade de Lisboa, Rua Teresa Ambrósio, 1600-277 Lisbon, Portugal; 4Center for Innovative Care and Health Technology (ciTechcare), Polytechnic of Leiria, 2410-541 Leiria, Portugal; 5School of Health Sciences, Polytechnic of Leiria, 2410-541 Leiria, Portugal; 6Nursing School, University of Évora, 7000-811 Évora, Portugal; 7Comprehensive Health Research Centre, University of Évora, 7000-811 Évora, Portugal; 8Nursing School of Porto, 4200-072 Porto, Portugal; 9Group Inovation and Development in Nursing (NursID), Centro de Investigação em Tecnologias e Serviços de Saúde (CINTESIS), 4200-450 Porto, Portugal

**Keywords:** mental health, confinement, addictive substances, higher education students

## Abstract

The mental health of higher education students is a constant concern, and the pandemic situation caused by COVID-19 has intensified this concern. The social measures imposed to control and minimize the disease have led, among other things, to the reconfiguration of higher education students’ academic life habits, which has naturally altered their emotional balance, mental health, and substance abuse. This cross-sectional, descriptive, and correlational study assesses the influence of higher education students’ personal characteristics on their (self-reported) use of addictive substances (alcohol, tobacco, drugs, and pharmaceutical drugs) before and during their first compulsory confinement in Portugal, as well as its relationship with mental health. An online questionnaire was applied between 15 April and 20 May 2020, to students from various study cycles of higher education institutions in one region of Portugal (northern area of Alentejo), which included the Mental Health Inventory in its reduced version (MHI-5) and questions (constructed by the authors) on personal characterization and on the use of addictive substances before and during confinement. The convenience sample included 329 mostly female health care students between the ages of 18 and 24. In our results, we found a statistically significant decrease in tobacco, alcohol, and drug use; however, there was an increase in tobacco use among older students and an increase in anxiolytic use among students with higher academic achievement and among students who exhibited more active social behavior in the period prior to confinement. Students who took anxiolytics during confinement had higher MHI-5 scores and students who used the most addictive substances during confinement had lower MHI-5 scores than the other students.

## 1. Introduction

At the beginning of the COVID-19 pandemic caused by SarsCov2, the WHO warned of the possibility of confinement with inevitable changes in life patterns and habits, some would become milder, others more intense. It also became predictable that mental health problems would increase in the population in general, and with students in particular, because there would be situations and constraints never experienced before and for which a very particular way of being was needed [1]. Depriving people of their freedom for the general good of the public is usually controversial and needs to be handled with care [2] because the psychological impact of quarantine can be broad, substantial, and long-lasting, but the psychological effects of not using it, and allowing the disease to spread can be worse [3]. Within this framework, measures of confinement and social distancing were adopted that immediately had a huge impact on education, affecting over 89.4% of higher education institutions in European countries [4].

The social measures used to combat the pandemic exacerbated the multiple risk factors, and we were all likely to feel anxiety due to the disruptions COVID-19 had on our daily lives, such as uncertainty about academic or professional future, loneliness, depression, and stress associated with worrying about the illness or death of family members and the inability to act in this new situation. These emotions are likely to have put us at increased risk for a range of unhealthy behaviors and coping strategies, including substance use and gambling [5].

Attending higher education is a stressful period for many students who experience the first onset of mental health and addictive substance-use problems or the exacerbation of their symptoms [6], showing increased rates of anxiety, depression, suicidal thoughts, trauma, and use of these substances [7]. They are more likely to report psychosocial distress compared to older adults and non-student youth in the general population [8].

During the pandemic, higher education institutions closed campuses, held online classes, and increased physical distancing. This caused substantial disruption to students’ lives, adding to the loss of family income, anxiety about future educational and employment prospects, and disconnection from the social interactions that are a normal part of college and young adult life [9]. Students reconfigured their lifestyles, reorganizing their daily activities, social interactions, and consumption, which were limited to what was possible, given their context and the greater or lesser accessibility of what they wanted.

A systematic review conducted by Layman et al. in 2022 on youth substance use during the COVID-19 pandemic found 49 studies, and most of them reported substantial reductions in prevalence across all categories of substance use (alcohol, tobacco, cannabis and e-cigarette) [10]; a fact that may be due to forced isolation, deprivation of contact with the group, and a context more conducive to substance access and consumption. No less important is the constant presence with the family that is an informal controller and inhibits certain consumptions [11], serving as a protective factor during the period of confinement. 

Other studies in Europe have shown that students who returned to their parents’ home during confinement were more likely to decrease their alcohol, tobacco, and cannabis use compared to students who lived continuously with their parents [12,13]; from which it can be inferred that living with peers in a more permissive rules context is associated with addictive substance use [14,15]. Students consume primarily and to a greater extent at social events [14,16]. Overall, there appeared to be less interest in drug use generally associated with recreational events [17]. 

Depressive symptoms were associated with a higher likelihood of increased use of all types of substances during confinement [12,13,18,19], including drugs by self-medication [20]. Poor mental health is a risk factor for substance use and abuse, and most youths with substance use problems suffer from concomitant mental health problems, which are often difficult to treat [21,22]. 

The average mental health of first-year university students worsened after the onset of the COVID-19 pandemic and recovered to the pre-pandemic level during the following two years [23]. 

In Portugal, the country where this study was conducted, confinement measures were also imposed, and the changes caused in higher education students’ life were noticeable. A study conducted by Maia and Dias in 2020 found a significant increase in psychological disturbances such as anxiety, depression, and stress among university students during the pandemic period compared to other periods [24]. A study by Vasconcelos et al., with a sample of Portuguese university students, found that in general students significantly decreased their alcohol consumption during confinement and post-confinement compared to the pre-COVID-19 period and that they stabilized at a low level of alcohol intake during the pandemic, despite an increase in alcohol cravings post-confinement [25]. A study conducted by Carneiro on a sample of young Portuguese men between 18 and 25 years old, found that during the first confinement most reported consuming much less alcohol and drugs than usual, but tobacco use remained the same [26,27].

In this framework and knowing that there are limited studies conducted in Portugal describing substance use by higher education students in periods of confinement, and making the association with their mental health, it is important to further research on the problem so that future interventions can be planned. The authors conducted a study in this particular region of the country, to enrich knowledge about the theme with the aim to assess the influence of personal characteristics on (self-reported) substance use (alcohol, tobacco, and drugs) before and during confinement and to analyze the association between mental health and (self-reported) substance use (alcohol, tobacco, and drugs) in higher education students of the northern area of Alentejo during confinement.

## 2. Materials and Methods

### 2.1. Study Design

This cross-sectional, descriptive, and correlational study used an online questionnaire administered on the Google^®^ Forms platform, applied from April to June 2020, to higher education students from a region of Portugal, during the first period of COVID-19 confinement. 

The following research questions were formulated:

(a) What is the association between the personal characteristics of higher education students in Portugal and (self-reported) substance use (alcohol, tobacco, and drugs) before and during confinement?

(b) What is the association between mental health and (self-reported) substance use (alcohol, tobacco, and drugs) in Portuguese higher education students during confinement?

Based on the research questions and objectives, the Mental Health Inventory-5 (MHI-5) was used to assess students’ mental health and the authors constructed (ad hoc) questions on personal characterization and (self-reported) substance use before and during the confinement. Mental health, “a state of well-being in which the individual realizes his or her own abilities, can cope with the normal stresses of life, can work productively and fruitfully, and is able to make a contribution to his or her community”, was considered as an independent variable [27]. Prevalence and type of substance use (alcohol, tobacco, and drugs) before and during confinement were considered to be independent variables.

### 2.2. Data Collection

A non-probabilistic convenience sample was obtained and calculated for a margin of error of 5% and a confidence level of 90%. The questionnaire was administered online via e-mail with the access link for completion and sent to 4450 students with active enrollment in the two higher education institutions in a region of Portugal. A total of 329 valid questionnaires were obtained, constituting our sample. 

At the beginning of the online questionnaire, the study was explained and the participant could only continue after giving his/her consent. Ethical issues were always safeguarded. Data confidentiality was guaranteed, and the collected data was stored in the researchers’ personal folders with security codes. The study was approved by the Ethics Committee of the Polytechnic Institute of Portalegre (Ethical Opinion no. SC/2020/316 of 20/02/2020) and by the data protection officers of both institutions. The procedures were conducted in accordance with the Helsinki Declaration.

The questionnaire of sociodemographic characterization of students was developed by the research team with questions (Ad-hoc) and integrates 12 self-report items: sex; age; marital status; type of love relationship; level of education; course; academic year; school performance rating; social life during the school period (parties and festivals); membership in an academic group; membership in a sports, recreational or political group; and membership in a religious, charitable or voluntary group.

The student characterization questionnaire regarding their consumption of addictive substances before and during COVID-19 was also developed by the research team with (Ad-hoc) questions and integrates nine self-report items on consumption: tobacco; wine or beer; distilled spirits; cannabis; cocaine; stimulants; opiates; anxiolytics; and sedatives.

Mental health was also assessed using the Mental Health Inventory in a reduced version (MHI-5) translated and validated for Portugal by Pais-Ribeiro in 2001 [28], and is composed of five items representing four dimensions of mental health: anxiety; depression; loss of emotion-behavioral control; and psychological well-being [29]. The rating of the scale is obtained by summing the items (two items with the rating reversed). Higher levels in the sum correspond to better mental health between five and thirty.

Before data collection, a pre-test of the questionnaire was applied to ten (10) students who did not integrate the sample, serving as a “dress rehearsal” for its application, and allowing the authors to verify the correctness and adequacy of the Ad hoc questions. After its application and analysis, no suggestions for change were reported, and the initial version was kept; it was considered adjusted to the objectives and easy to understand by the students invited to participate.

### 2.3. Statistical Analysis

Descriptive statistics (absolute and relative frequency, mean, and standard deviation) were used according to the type of variable to characterize the sample under study. To measure the association between two quantitative variables, a correlation was used; in the cases of ordinal variables or when distributions departed from normality, Spearman’s coefficient was used. To evaluate the strength of the association, intensity levels were used according to Marôco [30]: low correlation (0.21 to 0.39); moderate (0.41 to 0.69); high (0.71 to 0.89); very high >0.90. The chi-square test was used to compare the proportions between the study variables and the demographic characteristics analyzed. Data analysis was performed using the computer program SPSS version 27 with a significance level of 5%. 

## 3. Results

### 3.1. Sociodemographic Characteristics

The sample included 329 students from various masters, undergraduate and technical courses (Management, Nursing, Agronomy, Computer Engineering, Multimedia Product Development, Social Work, Gerontology, Occupational Therapy, Veterinary Nursing, Viticulture and Oenology, Agricultural Production, Accounting, Communication Design, Journalism and Communication, and Equine). The most representative characteristics are as follows: 80.5% were women; 82.7% were between 18 and 24 years old; 63.3% were single; 83.3% were attending a degree course; 66.9% rated their academic results as good; 26.1% attended one or two academic parties and festivals during the semester; 81.1% did not belong to any academic group or association; 84.8% did not belong to any religious, social solidarity, or volunteer group; 89.4% did not belong to any sports, recreational, or political group. Overall, they had an average of 19.78 for the MHI-5.

In our results, we can see that there were no students consuming cocaine or opioids and that the most consumed substance was alcohol. During the lock-in, the consumption of all substances decreased compared to the previous period, the first semester. The biggest decrease in users during the period of confinement was the users of alcohol, both wine and beer, and distilled beverages. It was also observed that those who consumed in larger quantities, in general, had lower MHI-5 values, as shown in Table 1.

For tobacco and cannabis use there are statistically significant differences (*p* < 0.001) when comparing before and after confinement, with a reduction in the number of cigarettes smoked and cannabis use. A statistical difference was also found in the use of anxiolytics, sedatives, or hypnotics (*p* < 0.001), with an increase in its consumption during confinement. For alcohol intake (wine or beer and distilled beverages), a statistically significant reduction in consumption was found (*p* < 0.001) during confinement.

When analyzing substance use during confinement and mental health (MHI-5), there is no statistical significance between mental health and tobacco use (*p* = 0.132) or cannabis use (*p* = 0.627). Statistically significant differences were found in anxiolytics use (*p* < 0.001), with a lower MHI-5 for higher consumption of anxiolytics; for wine or beer consumption (*p* < 0.001), with a higher MHI-5 for elevated consumption of wine or beer during confinement; and for distilled beverages (*p* < 0.001), with a lower MHI-5 for higher consumption.

This study’s results are also in line with what was previously mentioned about the main places of consumption. It was found that the most commonly mentioned places for consumption are bars or discos during group outings, as shown in Table 2.

Considering that the substance most consumed by students is alcohol, the effects of this consumption (self-reported) before and during confinement were identified. Results show that there were fewer users and lower consumption of alcohol, so the associated effects also decreased during the confinement, as presented in Table 3.

### 3.2. Association between Sex and Personal Characteristics and Substance Use

As noted above, in general, students decreased their substance consumption during confinement in parallel with their usual consumption; with the exception of anxiolytic consumption, which intensified during confinement. 

Considering the association between sex and personal characteristics of higher education students in Portugal and substance consumption (alcohol, tobacco, and drugs) before and during confinement, data are presented by sex and personal characteristics. For wine or beer, men consumed more than women before confinement (*p* = 0.019) and during confinement (*p* < 0.001). Participants over 44 years of age increased tobacco consumption during confinement (*p* = 0.020). For adequate and good grades, there was an increase in anxiolytic consumption (*p* < 0.001) during confinement (Table 4). 

For cocaine, stimulant, and opiate use, the small number or absence of responses does not allow for a conclusion.

### 3.3. Considering the Association between Mental Health and Substance Use in Higher Education in Portugal during Confinement

For mental health and tobacco, there is a weak negative correlation (ρ = −0.162) that is statistically significant (*p* = 0.003). The interpretation of this correlation indicates that the higher the tobacco use, the lower the mental health. 

For mental health and anxiolytics, there is a weak negative correlation (ρ = −0.257) that is statistically significant (*p* < 0.001). The interpretation of this correlation indicates that the higher the anxiolytic consumption, the lower the mental health. 

For all other variables analyzed, statistically significant differences in consumption were never found in any of the periods. 

## 4. Discussion

In this study it was observed that men consumed more alcohol than women, before and during confinement; however, there are significant differences related to wine or beer consumption, since it decreased during confinement for both sexs. It was noted that the consumption of distilled beverages by women also decreased during confinement; men maintained the same consumption level for distilled beverages. The observed decrease in alcohol consumption can be justified by the difficulty of access to the substance or by the decrease in social contact that could promote its consumption. The same conclusion is suggested by several studies mentioning that confinement was a protective factor against the consumption of alcohol (*p* < 0.001) [31,32]. In 2021, Busse carried out a study involving 5021 students and concluded that during the pandemic there was a change in the pattern of substance use. In addition, 61% of students reported consuming alcohol and 45.8% drinking alcohol in excess, but 24.4% reported a decrease in excessive alcohol consumption [18]. The lack of social interactions is also mentioned by Jackson in 2021, who conducted two studies in the USA involving 350 college students; in both studies, COVID-19-related alcohol consumption was accompanied by reductions in quantity, heavy drinking, and drunkenness for most students, and there was evidence of reductions in social drinking with friends and roommates and at parties. Alcohol consumption decreased due to reduced opportunities and/or social environments, limited access to alcohol, and reasons related to health and self-discipline [33]. Concerning age, a decrease was prevalent and significant, in our case, between the ages of 18 and 30 and 36 and 44. The same was found by Villanueva-Blasco in 2021, who found that although alcohol consumption during confinement showed a significant general decline, age revealed important differences, with the decline being more pronounced in adults from 18 to 29 years old [34]. Unlike this author, who found an increase in alcohol consumption in those who lived alone or with a partner, we found no increase in consumption related to marital status or type of romantic relationship.

Concerning tobacco consumption, in the present study men and women smoked less during confinement. For most characterization variables we found a reduction of consumption during confinement, except for the older participants (over 44 years old), who increased tobacco consumption. This reduction was not found by Busse, in 2021, in a study concluded during the pandemic period where a change in the pattern of substance use was found but the use of tobacco and cannabis remained the same [18]. Ruiz-Zaldibar mentioned in his study that confinement was a protective factor against the consumption of tobacco [31]. When considering the relationship type, a reduction in tobacco consumption was observed for the majority of types. A study conducted by Sokolovsky in 2021, involved 429 higher education students and concluded that although collegiate smokers and vapers decreased tobacco use frequency in response to campus closure, the quantity remained stable, suggesting that those with sustained tobacco use smoked and/or vaped more heavily on use days [35].

The use of cannabis in the present work diminished for every variable under study; the same was not observed in a study conducted in Spain where a higher risk of cannabis use was found in late adolescents (young adults) [36]. In addition, Daigre, in 2022, reported that younger people more frequently mention consuming cannabis during the COVID-19 lockdown [37]. The same decrease in cannabis consumption was seen in all variables related to the participants’ social life; this could be explained by the abrupt decrease in socialization opportunities with peers, which clearly provides and facilitates opportunities for consumption. This condition is also mentioned by Rogés J in 2021 [37]. A similar situation to the one experienced during the COVID-19 pandemic period occurred during 2002–2004 (SARS), where restrictions on social contacts were also experienced with negative effects of quarantine measures on the social participation of individuals [38,39].

In the present study, there was an increase in the consumption of anxiolytics in participants who had a higher level of academic results as well as in those who indicated belonging to one or two social groups related to academic activities. This may be due to increased anxiety related to the methodological changes of distance learning and consequences in academic results, and also the absence of social contact. In 2020, Son carried out a study in the USA involving 195 university students, of which 138 (71%) indicated an increase in stress and anxiety due to the COVID-19 outbreak, and identified stressors such as increased concerns about school performance [40]. Studies conducted in Europe, including Portugal, have shown very similar results. During confinement, students showed higher levels of anxiety [41,42]. 

In the study of mental health, it was observed in the present work that there is a negative correlation between tobacco and anxiolytic use with the mental health of the participants. This finding is supported by most literature that mentions that people with mental health conditions consume 44% of all cigarettes in western countries [43]. In addition, two in three of those individuals with severe mental health conditions are current smokers, and smoke approximately double that of the general population [44]

The results obtained in this study allow us to contribute to the formulation of recommendations related to students’ mental health and addictive substance use. The fact that an exceptional situation such as the one that occurred during confinement was experienced allows us to understand in a more in-depth way the conditions for the consumption of these types of substances related to students´ mental health.

As limitations, we can mention that this study presents results from a specific population of a district of Portugal and higher education students, and also that the sample is not representative of these students in terms of the sex of the respondents, which may be a weakness in the results obtained. It would be pertinent to develop similar studies at a national level, involving other population groups who also suffered from the limitations of confinement.

## 5. Conclusions

In this study, Portuguese college students from the region of northern Alentejo, reported reduced substance use during confinement. This can be explained by the difficulty of gaining access to these substances, and the presence of the family may also have been an inhibiting factor for the consumption of certain substances. The association between gender and personal characteristics of students in the northern Alentejo region and substance consumption (alcohol, tobacco, and drugs) before and during confinement indicates that men consumed more wine or beer than women before and during confinement. Tobacco consumption increased during confinement in participants over 44 years of age, and students who self-reported having adequate and good grades increased their consumption of anxiolytics during this same period. During the period of confinement, the consumption of all substances decreased compared to the period before confinement. The greatest decrease in consumption of addictive substances occurred in alcoholic beverages in general. Participants with lower MHI-5 values were those who resorted more frequently to the consumption of addictive substances.

The few cases in which an increase in consumption was observed seem to be clearly related to this confinement, such as the increase in tobacco use among older students and the increase in the use of anxiolytics among students with better academic results, who possibly feared a decrease in these results due to pedagogical changes and the interruption of face-to-face teaching sessions, and students who demonstrated more active social behavior in the period prior to confinement and who suddenly had to change their habits in terms of how they experienced the academic environment. 

## Figures and Tables

**Table 1 healthcare-11-00619-t001:** Addictive substance use before and during confinement, and mental health during confinement.

Additive Substance Use	Consumption Period	Consumption Type	Students		
			*n*	%	MHI-5
Tobacco	Before confinement	Does not smoke	221	67.1	
		Smokes sporadically	40	21.2	
		Smokes 1 to 4 cigarettes a day	16	4.8	
		Smokes 5 to 10 cigarettes a day	36	10.9	
		Smokes more than 10 cigarettes a day	16	4.8	
	During confinement	Does not smoke	259	78.7	19.8
		Smokes sporadically	29	8.8	18.9
		Smokes 1 to 4 cigarettes a day	15	4.5	20.6
		Smokes 5 to 10 cigarettes a day	17	5.1	19.8
		Smokes more than 10 cigarettes a day	9	2.7	18.5
Cannabis (hashish, weed, marijuana)	Before confinement	No consumption	301	91.4	
		I consume rarely	11	3.3	
		Sometimes	10	3.0	
		I consume every day	2	0.6	
		Frequently	5	1.5	
	During confinement	No consumption	317	96.3	19.7
		I consume rarely	7	2.1	20.1
		Sometimes	2	0.6	18
		I consume every day	0	0	
		Frequently	3	0.9	18
Cocaine (coke and crack)	Before confinement	No consumption	329	100	
	During confinement	No consumption	329	100	19.7
Stimulants (speed, amphetamines, ecstasy)	Before confinement	No consumption	325	98.7	
		I consume rarely	2	0.6	
		Sometimes	2	0.	
	During confinement	No consumption	329	100	19.5
Opioids (heroin, morphine, methadone)	Before confinement	No consumption	329	100	
	During confinement	No consumption	329	100	19.7
Anxiolytics, sedatives or hypnotics (pharmaceutical products)	Before confinement	No consumption	293	89.3	
		I consume rarely	14	4.3	
		Sometimes	11	3.4	
		I consume every day	8	2.4	
		Frequently	2	0.6	
	During confinement	No consumption	301	91.5	19.7
		I consume rarely	8	2.4	19.5
		Sometimes	9	2.7	19.4
		I consume every day	7	2.1	22
		Frequently	4	1.2	19.2
Wine or beer	Before confinement	No consumption	103	31.3	
		I consume rarely	73	22.1	
		Sometimes	98	29.7	
		I consume every day	8	24.3	
		Frequently	47	14.2	
	During confinement	No consumption	199	60.4	20.0
		I consume rarely	77	23.4	18.4
		Sometimes	41	12.4	21.0
		I consume every day	2	0.6	17.0
		Frequently	10	3.0	19.8
Distilled beverages (shots, whisky, liqueurs, brandy)	Before confinement	No consumption	122	37.0	
		I consume rarely	100	30.3	
		Sometimes	84	25.5	
		I consume every day	0	0	
		Frequently	23	6.9	
	During confinement	No consumption	252	76.5	19.7
		I consume rarely	55	16.7	20.6
		Sometimes	19	5.7	18.7
		I consume every day	0	0	
		Frequently	3	0.9	18.3

**Table 2 healthcare-11-00619-t002:** Main place of substance consumption before confinement.

Main Place of Substance Consumption before Confinement	Students	
	*n* = 329	%
Not applicable	97	29.4
Academic parties or festivals	59	17.9
Any place or occasion	34	10.3
Outings (in the late afternoon or evening) to bars or nightclubs	138	41.9
Alone at home	5	1.5

**Table 3 healthcare-11-00619-t003:** Effects of alcohol consumption before and during confinement.

Effects of Alcohol Consumption		before Confinement		during Confinement		
		*n* = 329	%	*n* = 329	%	MHI-5
Drinks alcoholic beverages until he gets drunk	Not applicable	151	45.8	298	9.0	19.9
	Rarely	81	24.6	21	6.3	18.2
	Sometimes	70	21.2	8	2.4	18.8
	Frequently	19	5.7	0	0	-
	Almost always	8	2.4	2	0.6	21.5
Drinks alcoholic beverages until he gets drunk and loses track of his attitudes	Not applicable	269	81.7	319	96.9	19.8
	Rarely	46	13.9	9	2.7	18.2
	Sometimes	11	3.3	1	0.3	22
	Frequently	1	0.3	0	0	-
	Almost always	2	0.6	0	0	-
When you drink too much and have trouble concentrating in class the next day	Not applicable	190	57.7	304	92.4	19.8
	Rarely	75	22.7	16	4.8	19.1
	Sometimes	44	13.3	7	2.1	19
	Frequently	12	3.6	1	0.3	24
	Almost always	8	2.4	1	0.3	24

**Table 4 healthcare-11-00619-t004:** Association between sex and personal characteristics and substance use (Pearson chi-square test).

Sex and Personal Characteristics		Alcohol Decrease before vs. during	Tobacco Decrease before vs. during	Cannabis Decrease before vs. during	Anxiolytics Decrease before vs. during
Sex	Female Male	*p* < 0.001 *p* = 0.004	*p* < 0.001 *p* < 0.001	*p* < 0.001 *p* < 0.001	*p* < 0.001 *p* = 0.011
Age	18 to 24 25 to 30 31 to 35 36 to 44 >44	*p* = 0.001 *p* = 0.001 *p* > 0.050 *p* = 0.013 *p* > 0.050	*p* < 0.001 *p* < 0.001 *p* < 0.019 *p* < 0.001 *p* > 0.050	*p* < 0.001 *p* = 0.013 *p* > 0.050 *p* = 0.007 *p* > 0.050	*p* > 0.050 *p* = 0.008 *p* = 0.014 *p* = 0.007 *p* > 0.050
Marital/Relational status	Married/civil union Single/divorced No relationship Widowed	*p* = 0.047 *p* < 0.001 *p* < 0.001 no cases	*p* < 0.005 *p* < 0.005 *p* < 0.005 no cases	*p* > 0.050 *p* < 0.001 *p* < 0.001 no cases	*p* = 0.003 *p* < 0.001 *p* < 0.001 no cases
Love/Relationship type	Not applicable Long and non-conflictive (+6 months) Occasional, short and non-conflictive Long and conflictive (+6 months) Occasional, short and conflictive	*p* < 0.001 *p* < 0.001 *p* > 0.005 *p* > 0.050 *p* > 0.050	*p* < 0.001 *p* < 0.001 *p* = 0.007 *p* > 0.050 *p* > 0.050	*p* < 0.001 *p* < 0.001 *p* > 0.050 *p* = 0.034 *p* > 0.050	*p* < 0.001 *p* < 0.001 *p* > 0.050 *p* > 0.050 *p* = 0.005
Academic Classification	Mediocre Adequate Good Very Good	*p* > 0.050 *p* < 0.001 *p* < 0.001 *p* = 0.007	*p* > 0.050 *p* < 0.001 *p* < 0.001 *p* < 0.001	*p* > 0.050 *p* < 0.001 *p* < 0.001 *p* > 0.050	*p* > 0.050 *p* < 0.001 *p* < 0.001 *p* > 0.050
Social life—Integrates a religious group, social solidarity or volunteer work	Not Applicable 1 group 2 groups 3 groups 4 groups or +	*p* < 0.001 *p* = 0.025 *p* = 0.018 *p* > 0.050 *p* > 0.050	*p* < 0.001 *p* < 0.001 *p* = 0.034 *p* > 0.050 *p* > 0.050	*p* < 0.001 *p* > 0.050 *p* > 0.050 *p* > 0.050 *p* > 0.050	*p* < 0.001 *p* = 0.023 *p* = 0.010 *p* > 0.050 *p* > 0.050
Social life—member of a sports, recreational, or political group	Not Applicable 1 group 2 groups 3 groups 4 groups or +	*p* < 0.001 *p* = 0.005 *p* > 0.050 *p* > 0.050 *p* > 0.050	*p* < 0.005 *p* < 0.005 *p* = 0.019 *p* > 0.050 *p* > 0.050	*p* < 0.001 *p* = 0.043 *p* > 0.050 *p* > 0.050 *p* > 0.050	*p* < 0.001 *p* = 0.010 *p* > 0.050 *p* > 0.050 *p* > 0.050
Attended academic parties and festival before confinement	Does not attend Attends 1 to 2 Attends 3 to 5 Attends 6 to 10 +10	*p* < 0.001 *p* < 0.001 *p* = 0.002 *p* = 0.030 *p* > 0.050	*p* < 0.001 *p* < 0.001 *p* < 0.001 *p* < 0.001 *p* < 0.001	*p* < 0.001 *p* = 0.004 *p* = 0.021 *p* > 0.050 *p* < 0.001	*p* < 0.005 *p* < 0.005 *p* < 0.005 *p* < 0.005 *p* < 0.005
Member of academic group before confinement	Not Applicable 1 group 2 groups 3 groups 4 groups or +	*p* < 0.001 *p* = 0.020 *p* > 0.050 *p* > 0.050 *p* > 0.050	*p* < 0.001 *p* < 0.001 *p* = 0.037 *p* > 0.050 *p* > 0.050	*p* < 0.001 *p* = 0.002 *p* > 0.050 *p* > 0.050 *p* < 0.034	*p* < 0.001 *p* > 0.050 *p* > 0.050 *p* > 0.050 *p* > 0.050

## Data Availability

Data is available on request due to ethical restrictions.

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
