# Peer review of "The Impact of COVID-19 Confinement on Substance Use and Mental Health in Portuguese Higher Education Students"

_healthcare, 2023, doi:10.3390/healthcare11040619_

Round 1

Reviewer 1 Report

Dear Authors,

Thank you for the opportunity to review your manuscript. I was eager to learn about the results of your work. The topic fits in with my interests. Below are my suggestions:

- the theoretical introduction is too long, please make it shorter and thematically unified.

- Part 2.1. please move to the target. This is a premise, not a methodological part.

- Is the study group homogeneous and sociodemographically representative? If not then please add this to the limitations.

- the tables lack references to test scores and correlation coefficients.

- at the end of the discussion, please add a paragraph about the strengths and limitations of the work (one I have already pointed out!).

- the conclusions are too long and not synthetic enough. Please present them in short paragraphs.

Author Response

Dear Reviewer 1,

The authors appreciate and recognize the work done by the reviewer 1 and express their gratitude for the comments. The answers to your comments are as follows:

 - the theoretical introduction is too long, please make it shorter and thematically unified.

We have shortened the introduction, as suggested. It now reads easier, in our opinion.

- Part 2.1. please move to the target. This is a premise, not a methodological part.

We change it according to your suggestion.

- Is the study group homogeneous and sociodemographically representative? If not then please add this to the limitations.

No, it is not, it is a convenience sample from Higher Education Institutions that do not represent the country. We added this information to the study limitations, it now reads: As limitations we can mention that this study presents results from a specific population of a district of Portugal and higher education students, and also that the sample is not representative of these students in terms of the gender of the respondents, which may be a weakness in the results obtained. It would be pertinent to develop similar studies at national level involving other population groups who also suffered the limitations of confinement.

- the tables lack references to test scores and correlation coefficients.

The reference was introduced, and it is now in table 4 title, it reads: Table 4. Association between sex and personal characteristics and substance use (Pearson chi-square test) .

- at the end of the discussion, please add a paragraph about the strengths and limitations of the work (one I have already pointed out!).

We added the paragraph at the end of discussion as suggested, it now reads: As limitations we can mention that this study presents results from a specific population of a district of Portugal and higher education students, and also that the sample is not representative of these students in terms of the gender of the respondents, which may be a weakness in the results obtained. It would be pertinent to develop similar studies at national level involving other population groups who also suffered the limitations of confinement.

- the conclusions are too long and not synthetic enough. Please present them in short paragraphs.

We have changed the conclusion, as suggested, making it more focused to answer the study questions.

Reviewer 2 Report

The authors’ study is to evaluate the influence of higher education students' personal characteristics on their sub-stances (alcohol, tobacco, drugs, and pharmaceutical drugs) usage before and during their first compulsory confinement in Portugal. The topic is interesting. I would like to give the following comments.

1.     If the authors did case study among a particular area in Portugal, it’s better to appear in the title.

2.     End of introduction should be explained research gaps based on previous studies.

3.     From the beginning I can answer the research questions, so please bring more evidence why we need such research.

4.     How did the authors measure the sample size?

5.     Their questionnaire was designed based on English. If It’s not. How did they validate it?

6.     There is no clear evidence for measuring research variables. It’s better to prepare a table that shows questions in every research variable. Specialty MHI-5

  1. Explain more about questionnaire design. Is there any pilot study?
  2. Please prepare the reliability and validity of your questionnaire.
  3. I would like to see their analysis regarding their missing data and outliers.
  4. It’s better to consider a Table to show the results of their hypotheses.
  5. Why authors didn’t use “fear Covid-19” and its impact on their main research variables.

12.  Contribution of the study is not strong enough. What is the novelty of their study?

Author Response

Dear Reviewer 2,

The authors appreciate and recognize the work done by the reviewer 2 and express their gratitude for the comments. The answers to your comments are as follows:

The authors’ study is to evaluate the influence of higher education students' personal characteristics on their sub-stances (alcohol, tobacco, drugs, and pharmaceutical drugs) usage before and during their first compulsory confinement in Portugal. The topic is interesting. I would like to give the following comments.

If the authors did case study among a particular area in Portugal, it’s better to appear in the title.

We have changed the title according to your suggestion. It now reads: The Impact of COVID-19 confinement on Substance Use and Mental Health in Portuguese Higher Education Students

End of introduction should be explained research gaps based on previous studies.

We did add some more information on the work done to fill these gaps in knowledge at the end of introduction. It now reads: In this framework and knowing that there are limited studies conducted in Portugal describing substance use by higher education students in periods of confinement, and making the association with their mental health, it is important to further research on the problem so that future interventions can be planned. The authors conducted a study in this particular region of the country, to enrich knowledge about the theme with the aim to assess the influence of personal characteristics on (self-reported) substance use (alcohol, tobacco and drugs) before and during confinement and to analyze the association between mental health and (self-reported) substance use (alcohol, tobacco and drugs) in higher education students of northern area of Alentejo during confinement.

From the beginning I can answer the research questions, so please bring more evidence why we need such research.

We have covered this subject in the previous paragraph, the specific region of Portugal where  this study was developed and also the use of the MHI-5 instrument justifies the need for this research.

How did the authors measure the sample size?

The sample size calculation is presented in lines 143-144 and it reads: A non-probabilistic convenience sample was obtained, calculated for a margin of error of 5% and a confidence level of 90%.

Their questionnaire was designed based on English. If It’s not. How did they validate it?

The questionnaire used in this study was translated and validated by a Portuguese author, other than the authors of this study, this is mentioned in the methods section (lines 166-171 and it reads: Mental health was also assessed using the Mental Health Inventory in a reduced version (MHI-5) translated and validated for Portugal by Pais-Ribeiro in 2001 [28] and is composed of five items representing four dimensions of mental health: Anxiety, Depression, Loss of Emotion-Behavioral Control, and Psychological Well-Being [29]. The rating of the scale is obtained by summing the items (2 items with the rating reversed). Higher levels in the sum correspond to better mental health between 5 and 30).

The other questionnaires developed by the research team (The Questionnaire of sociodemographic characterization of students 12 self-report items with questions (Ad-hoc) on: sex; age; marital status; type of love relationship; level of education; course; academic year; school performance rating; social life during the school period (parties and festivals); membership in an academic group; membership in a sports, recreational or political group; membership in a religious, charitable or voluntary group. The student characterization questionnaire regarding their consumption of addictive substances before and during COVID-19 was also developed by the research team with nine self-report items (Ad-hoc) on the consumption of: tobacco; wine or beer; distilled spirits; cannabis; cocaine; stimulants; opiates; anxiolytics; and sedatives.) are of sample characterization and were not translated from English.

There is no clear evidence for measuring research variables. It’s better to prepare a table that shows questions in every research variable. Specialty MHI-5

The authors apologize in advance if they have not understood your question, but believe that the desired answers are already found in tables 1 and 3.

Explain more about questionnaire design. Is there any pilot study?

The sample characterization questionnaires are just for collection of the self-reported information from the participants and there was no pilot study for it. The MHI-5 was developed by another author and the information is available at the reference indicated in the introduction (Pais Ribeiro, J. L. "Mental health inventory: Um estudo de adaptação à população portuguesa." Psicol. Saude Doenças (2001): 77-99.)

Please prepare the reliability and validity of your questionnaire.

Please see the answer to your previous question

I would like to see their analysis regarding their missing data and outliers.

Missing data were not analyzed as they correspond to consumption situations that participants did not mention as having occurred before or during confinement. As was the case with the use of cocaine or opioids.

It’s better to consider a Table to show the results of their hypotheses.

We have no study hypotheses as this is not an experimental study, however clearer information was added in the conclusions with answers to the study questions, where it now reads: In this study, Portuguese college students from the region of northern Alentejo, reported reduced substance use during confinement. This can be explained by the difficulty of access to these substances and the presence of the family may also be an inhibiting factor for the consumption of certain substances. The association between gender and personal characteristics of students in the northern Alentejo region and substance consumption (alcohol, tobacco and drugs) before and during confinement indicates that for wine or beer men consumed more than women before and during confinement. Tobacco consumption increased during confinement in participants over 44 years of age, and students who self-reported having adequate and good grades increased their consumption of anxiolytics during this same period. During the period of confinement, the consumption of all substances decreased compared to the period before confinement. The greatest decrease in consumption of addictive substances occurred in alcoholic beverages in general. Participants with lower MHI-5 values were those who resorted more frequently to the consumption of addictive substances. The few cases in which an increase in consumption was observed seem to be clearly related to this confinement, such as the increase in tobacco use among older students and the increase in the use of anxiolytics among students with better academic results, who possibly feared a decrease in these results due to pedagogical changes and the interruption of face-to-face teaching sessions, and students who demonstrated more active social behavior in the period prior to confinement and who suddenly had to change their habits more in the way they experienced the academic environment.

Why authors didn’t use “fear Covid-19” and its impact on their main research variables.

When we constructed the questionnaire we did not think to address this question, but it is a pertinent suggestion. Thank you very much for this alert.

Contribution of the study is not strong enough. What is the novelty of their study?

As we wrote in the answer to your second question: The authors conducted a study in this particular region of the country, to enrich knowledge about the theme with the aim to assess the influence of personal characteristics on (self-reported) substance use (alcohol, tobacco and drugs) before and during confinement and to analyze the association between mental health and (self-reported) substance use (alcohol, tobacco and drugs) in higher education students of northern area of Alentejo during confinement.

Round 2

Reviewer 2 Report

The authors amended all of my comments. Thank you!